# Statistical analysis of coupled time series with Kernel Cross-Spectral Density operators.

**Michel Besserve**
MPI for Intelligent Systems and MPI for Biological Cybernetics, Tübingen, Germany
`michel.besserve@tuebingen.mpg.de`

**Nikos K. Logothetis**
MPI for Biological Cybernetics, Tübingen
`nikos.logothetis@tuebingen.mpg.de`

**Bernhard Schölkopf**
MPI for Intelligent Systems, Tübingen
`bs@tuebingen.mpg.de`

## Abstract

Many applications require the analysis of complex interactions between time series. These interactions can be non-linear and involve vector valued as well as complex data structures such as graphs or strings. Here we provide a general framework for the statistical analysis of these dependencies when random variables are sampled from stationary time-series of arbitrary objects. To achieve this goal, we study the properties of the Kernel Cross-Spectral Density (KCSD) operator induced by positive definite kernels on arbitrary input domains. This framework enables us to develop an independence test between time series, as well as a similarity measure to compare different types of coupling. The performance of our test is compared to the HSIC test using i.i.d. assumptions, showing improvements in terms of detection errors, as well as the suitability of this approach for testing dependency in complex dynamical systems. This similarity measure enables us to identify different types of interactions in electrophysiological neural time series.

## 1   Introduction

Complex dynamical systems can often be observed by monitoring time series of one or more variables. Finding and characterizing dependencies between several of these time series is key to understand the underlying mechanisms of these systems. This problem can be addressed easily in linear systems [4], however non-linear systems are much more challenging. Whereas higher order statistics can provide helpful tools in specific contexts [15], and have been extensively used in system identification, causal inference and blind source separation (see for example [10, 13, 5]); it is difficult to derive a general approach with solid theoretical results accounting for a broad range of interactions. Especially, studying the relationships between time series of arbitrary objects such as texts or graphs within a general framework is largely unaddressed.

On the other hand, the dependency between independent identically distributed (i.i.d.) samples of arbitrary objects can be studied elegantly in the framework of positive definite kernels [19]. It relies on defining cross-covariance operators between variables mapped implicitly to Reproducing Kernel Hilbert Spaces (RKHS) [7]. It has been shown that when using a characteristic kernel for the mapping [9], the properties of RKHS operators are related to statistical independence between input variables and allow testing for it in a principled way with the Hilbert-Schmidt Independence Criterion (HSIC) test [11]. However, the suitability of this test relies heavily on the assumption that i.i.d. samples of random variables are used. This assumption is obviously violated in any non-trivial setting involving time series, and as a consequence trying to use HSIC in this context can lead to incorrect conclusions. Zhang et al. established a framework in the context of Markov chains

[22], showing that a structured HSIC test still provides good asymptotic properties for absolutely regular processes. However, this methodology has not been assessed extensively in empirical time series. Moreover, beyond the detection of interactions, it is important to be able to characterize the nature of the coupling between time series. It was recently suggested that generalizing the concept of cross-spectral density to Reproducible Kernel Hilbert Spaces (RKHS) could help formulate non-linear dependency measures for time series [2]. However, no statistical assessment of this measure has been established. In this paper, after recalling the concept of kernel spectral density operator, we characterize its statistical properties. In particular, we define independence tests based on this concept as well as a similarity measure to compare different types of couplings. We use these tests in section 4 to compute the statistical dependencies between simulated time series of various types of objects, as well as recordings of neural activity in the visual cortex of non-human primates. We show that our technique reliably detects complex interactions and provides a characterization of these interactions in the frequency domain.

## 2 Background and notations

**Random variables in Reproducing Kernel Hilbert Spaces**

Let $\mathcal{X}_1$ and $\mathcal{X}_2$ be two (possibly non vectorial) input domains. Let $k_1(.,.) : \mathcal{X}_1 \times \mathcal{X}_1 \to \mathbb{C}$ and $k_2(.,.) : \mathcal{X}_2 \times \mathcal{X}_2 \to \mathbb{C}$ be two positive definite kernels, associated to two separable Hilbert spaces of functions, $\mathcal{H}_1$ and $\mathcal{H}_2$ respectively. For $i \in \{1, 2\}$, they define a canonical mapping from $x \in \mathcal{X}_i$ to $\boldsymbol{x} = k_i(., x) \in \mathcal{H}_i$, such that $\forall f \in \mathcal{H}_i$, $f(x) = \langle f, \boldsymbol{x} \rangle_{\mathcal{H}_i}$ (see [19] for more details). In the same way, this mapping can be extended to random variables, so that the random variable $X_i \in \mathcal{X}_i$ is mapped to the random element $\mathbf{X}_i \in \mathcal{H}_i$. Statistical objects extending the classical mean and covariance to random variables in the RKHS are defined as follows:

- the Mean Element (see [1, 3]): $\boldsymbol{\mu}_i = \mathbb{E}\left[\mathbf{X}_i\right]$,
- the Cross-covariance operator (see [6]): $\mathbf{C}_{ij} = \text{Cov}\left[\mathbf{X}_i, \mathbf{X}_j\right] = \mathbb{E}[\mathbf{X}_i \otimes \mathbf{X}_j^*] - \boldsymbol{\mu}_i \otimes \boldsymbol{\mu}_j^*$,

where we use the tensor product notation $f \otimes g^*$ to represent the rank one operator defined by $f \otimes g^* = \langle g, . \rangle f$ (following [3]). As a consequence, the cross-covariance can be seen as an operator in $\mathcal{L}(\mathcal{H}_j, \mathcal{H}_i)$, the Hilbert space of linear Hilbert-Schmidt operators from $\mathcal{H}_j$ to $\mathcal{H}_i$ (isomorphic to $\mathcal{H}_i \otimes \mathcal{H}_j^*$). Interestingly, the link between $\mathbf{C}_{ij}$ and covariance in the input domains is given by the Hilbert-Schmidt scalar product

$$\left\langle \mathbf{C}_{ij}, f_i \otimes f_j^* \right\rangle_{HS} = \text{Cov}\left[f_i(\mathbf{X}_i), f_j(\mathbf{X}_j)\right], \quad \forall(f_i, f_j) \in \mathcal{H}_i \otimes \mathcal{H}_j$$

Moreover, the Hilbert-Schmidt norm of the operator in this space has been proved to be a measure of independence between two random variables, whenever kernels are characteristic [11]. Extension of this result has been provided in [22] for Markov chains. If the time series are assumed to be k-order Markovian, then results of the classical HSIC can be generalized for a structured HSIC using universal kernels based on the state vectors $(x_1(t), \ldots, x_1(t+k), x_2(t), \ldots, x_2(t+k))$. The statistical performance of this methodology has not been studied extensively, in particular its sensitivity to the dimension of the state vector. The following sections propose an alternative methodology.

**Kernel Cross-Spectral Density operator**

Consider a bivariate discrete time random process on $\mathcal{X}_1 \times \mathcal{X}_2 : \{(X_1(t), X_2(t))\}_{t \in \mathbb{Z}}$. We assume stationarity of the process and thus use the following translation invariant notations for the mean elements and cross-covariance operators:

$$\mathbb{E}\mathbf{X}_i(t) = \boldsymbol{\mu}_i, \quad \text{Cov}\left[\mathbf{X}_i(t+\tau), \mathbf{X}_j(t)\right] = \mathbf{C}_{ij}(\tau)$$

The cross-spectral density operator was introduced for stationary signals in [2] based on second order cumulants. Under mild assumptions, it is a Hilbert-Schmidt operator defined for all normalized frequencies $\nu \in [0\,;\,1]$ as:

$$\mathbf{S}_{12}(\nu) = \sum_{k \in \mathbb{Z}} \mathbf{C}_{12}(k)\exp(-k2\pi\nu) = \sum_{k \in \mathbb{Z}} \mathbf{C}_{12}(k)z^{-k}, \text{ for } z = e^{2\pi\mathbf{i}\nu}.$$

This object summarizes all the cross-spectral properties between the families of processes $\{f(X_1)\}_{f\in\mathcal{H}_1}$ and $\{g(X_2)\}_{g\in\mathcal{H}_2}$ in the sense that the cross-spectrum between $f(X_1)$ and $g(X_2)$ is given by $S_{12}^{f,g}(\nu) = \langle f, \mathbf{S}_{12}g\rangle$. We therefore refer to this object as the Kernel Cross-Spectral Density operator (KCSD).

## 3 Statistical properties of KCSD

**Measuring independence with the KCSD**

One interesting characteristic of the KCSD is given by the following theorem [2]:

**Theorem 1.** *Assume the kernels $k_1$ and $k_2$ are characteristic [9]. The processes $X_1$ and $X_2$ are **pairwise independent** (i.e. for all integers t and t', $X_1(t)$ and $X_2(t')$ are independent), if and only if $\left\|\mathbf{S}_{12}(\nu)\right\|_{HS} = 0,\ \forall\nu\in[0,\,1]$.*

While this theorem states that KCSD can be used to test pairwise independence between time series, it does not imply independence between arbitrary sets of random variables taken from each time series in general. However, if the joint probability distribution of the time series is encoded by a Directed Acyclic Graph (DAG), the following Theorem shows that independence in this broader sense is achieved under mild assumptions.

**Proposition 2.** *If the joint probability distribution of time series is encoded by a DAG with no confounder under the Markov property and faithfulness assumption, pairwise independence between time series implies the mutual independence relationship $\{X_1(t)\}_{t\in\mathbb{Z}} \perp\!\!\!\perp \{X_2(t)\}_{t\in\mathbb{Z}}$.*

*Proof.* The proof uses the fact that the faithfulness and Markov property assumptions provide an equivalence between the independence of two sets of random variables and the d-separation of the corresponding sets of nodes in the DAG (see [17]). We start by assuming pairwise independence between the time series.

For arbitrary times $t$ and $t'$, assume the DAG contains an arrow linking the nodes $X_1(t)$ and $X_2(t')$. This is an unblocked path linking this two nodes; thus they are not d-separated. As a consequence of faithfulness, $X_1(t)$ and $X_2(t')$ are not independent. Since this contradicts our initial assumptions, there cannot exist any arrow between $X_1(t)$ and $X_2(t')$.

Since this holds for all $t$ and $t'$, there is no path linking the nodes of each time series and we have $\{X_1(t)\}_{t\in\mathbb{Z}} \perp\!\!\!\perp \{X_2(t)\}_{t\in\mathbb{Z}}$ according to the Markov property (any joint probability distribution on the nodes will factorize in two terms, one for each time series). $\qquad\square$

As a consequence, the use of KCSD to test for independence is justified under the widely used faithfulness and Markov assumptions of graphical models. As a comparison, the structured HSIC proposed in [22] is theoretically able to capture all dependencies within the range of k samples by assuming k-order Markovian time series.

**Fourth order kernel cumulant operator**

Statistical properties of KCSD require assumptions regarding the higher order statistics of the time series. Analogously to covariance, higher order statistics can be generalized as operators in (tensor products of) RKHSs. An important example in our setting is the joint quadricumulant (4th order cumulant) (see [4]). We skip the general expression of this cumulant to focus on its simplified form for four centered scalar random variables:

$$\kappa(X_1, X_2, X_3, X_4) = \mathbb{E}[X_1 X_2 X_3 X_4] - \mathbb{E}[X_1 X_2]\mathbb{E}[X_3 X_4] - \mathbb{E}[X_1 X_3]\mathbb{E}[X_2 X_4]$$
$$- \mathbb{E}[X_1 X_4]\mathbb{E}[X_2 X_3] \quad (1)$$

This object can be generalized to the case random variables mapped in two RKHSs. The quadricumulant operator $\mathcal{K}_{1234}$ is a linear operator in the Hilbert space $\mathcal{L}(\mathcal{H}_1 \otimes \mathcal{H}_2^*, \mathcal{H}_1 \otimes \mathcal{H}_2^*)$, such that $\kappa(f_1(X_1), f_2(X_2), f_3(X_3), f_4(X_4)) = \langle f_1 \otimes f_2^*, \mathcal{K}_{1234} f_3 \otimes f_4^*\rangle$, for arbitrary elements $f_i$. The properties of this operator will be useful in the next sections due to the following lemma.

**Lemma 3.** *[Property of the tensor quadricumulant] Let $\mathbf{X}_1^c, \mathbf{X}_3^c$ be centered random elements in the Hilbert space $\mathcal{H}_1$ and $\mathbf{X}_2^c, \mathbf{X}_4^c$ centered random elements in $\mathcal{H}_2$ (the centered random element is defined by $\mathbf{X}_i^c = \mathbf{X}_j - \boldsymbol{\mu}_j$), then*

$$\mathbb{E}\left[\langle \mathbf{X}_1^c, \mathbf{X}_3^c\rangle_{\mathcal{H}_1}\langle \mathbf{X}_2^c, \mathbf{X}_4^c\rangle_{\mathcal{H}_2}\right] = \operatorname{Tr}\mathcal{K}_{1234} + \langle C_{1,2}, C_{3,4}\rangle + \operatorname{Tr}C_{1,3}\operatorname{Tr}C_{2,4} + \langle C_{1,4}, C_{3,2}\rangle$$

In the case of two jointly stationary time series, we define the translation invariant quadricumulant between the two stationary time series as:

$$\mathcal{K}_{12}(\tau_1, \tau_2, \tau_3) = \mathcal{K}_{1234}(\mathbf{X}_1(t+\tau_1), \mathbf{X}_2(t+\tau_2), \mathbf{X}_1(t+\tau_3), \mathbf{X}_2(t))$$

**Estimation with the Kernel Periodogram**

In the following, we address the problem of estimating the properties of cross-spectral density operators from finite samples. The idea for doing this analytically is to select samples from a time-series with a tapering window function $w : \mathbb{R} \mapsto \mathbb{R}$ with a support included in $[0, 1]$. By scaling this window according to $w_T(k) = w(k/T)$, and multiplying it with the time series, T samples of the sequence can be selected. The windowed *periodogram* estimate of the KCSD operator for T successive samples of the time series is

$$\mathbf{P}_{12}^T(\nu) = \frac{1}{T\|w\|^2}\mathcal{F}_T[\mathbf{X}_1^c](\nu) \otimes \mathcal{F}_T[\mathbf{X}_2^c](\nu)^*,$$

$$\text{with } \mathbf{X}_i^c(k) = \mathbf{X}_i(k) - \boldsymbol{\mu}_i \text{ and } \|w\|^2 = \int_0^1 w^2(t)dt$$

where $\mathcal{F}_T[\mathbf{X}_1^c] = \sum_{k=1}^{T} w_T(k)(\mathbf{X}_1^c(k))z^{-k}$, for $z = e^{2\pi \mathbf{i}\nu}$, is the windowed Fourier transform of the delayed time series in the RKHS. Properties of the windowed Fourier transform are related to the regularity of the tapering window. In particular, we will chose a tapering window of bounded variation. In such a case, the following lemma holds (see supplementary material for the proof).

**Lemma 4.** *[A property of bounded variation functions] Let $w$ be a bounded function of bounded variation then for all $k$, $\left|\sum_{t=-\infty}^{+\infty} w_T(t+k)w(t) - \sum_{t=-\infty}^{+\infty} w_T(t)^2\right| \leq C|k|$*

Using this assumption, the above periodogram estimate is asymptotically unbiased as shown in the following theorem

**Theorem 5.** *Let $w$ be a bounded function of bounded variation, if $\sum_{k\in\mathbb{Z}} |k|\|\mathbf{C}_{12}(k)\|_{HS} < +\infty$, $\sum_{k\in\mathbb{Z}} |k| \operatorname{Tr}(\mathbf{C}_{ii}(k)) < +\infty$ and $\sum_{(k,i,j)\in\mathbb{Z}^3} \left|\operatorname{Tr}[\mathcal{K}_{12}(k,i,j)]\right| < +\infty$,*

$$\text{then } \lim_{T\to+\infty} \mathbb{E}\,\mathbf{P}_{12}^T(\nu) = \mathbf{S}_{12}(\nu), \; \nu \not\equiv 0 \pmod{1/2}$$

*Proof.* By definition,

$$\mathbf{P}_{12}^T(z) = \frac{1}{T\|w\|^2}\left(\sum_{k\in\mathbb{Z}} w_T(k)\mathbf{X}_1^c(k)z^{-k}\right) \otimes \left(\sum_{n\in\mathbb{Z}} w_T(n)\mathbf{X}_2^c(n)z^{-n}\right)^*$$

$$= \frac{1}{T\|w\|^2}\sum_{k\in\mathbb{Z}}\sum_{n\in\mathbb{Z}} z^{n-k} w_T(k)w_T(n)\mathbf{X}_1^c(k) \otimes \mathbf{X}_2^c(n)^*$$

$$= \frac{1}{T\|w\|^2}\sum_{\delta\in\mathbb{Z}} z^{-\delta}\sum_{n\in\mathbb{Z}} w_T(n+\delta)w_T(n)\mathbf{X}_1^c(n+\delta) \otimes \mathbf{X}_2^c(n)^*, \text{ using } \delta = k - n.$$

Thus using Lemma 4,

$$\mathbb{E}\,\mathbf{P}_{12}^T(z) = \frac{1}{T\|w\|^2}\sum_{\delta\in\mathbb{Z}} z^{-\delta}(\sum_{n\in\mathbb{Z}} w_T(n)^2 + O(|\delta|))\mathbf{C}_{12}(\delta)$$

$$= \frac{1}{\|w\|^2}(\sum_{n\in\mathbb{Z}} \frac{w_T(n)^2}{T})\sum_{\delta\in\mathbb{Z}} z^{-\delta}\mathbf{C}_{12}(\delta) + \frac{1}{T}O(\sum_{\delta\in\mathbb{Z}} |\delta|\|\mathbf{C}_{12}(\delta)\|_{HS}) \xrightarrow[T\to+\infty]{} \mathbf{S}_{12}. \quad\square$$

However, the squared Hilbert-Schmidt norm of $\mathbf{P}_{12}^T(\nu)$ is an asymptotically biased estimator of the population KCSD squared norm according to the following theorem.

**Theorem 6.** *Under the assumptions of Theorem 5, for $\nu \not\equiv 0 \pmod{1/2}$*

$$\lim_{T\to+\infty} \mathbb{E}\left\|\mathbf{P}_{12}^T(\nu)\right\|_{HS}^2 = \left\|\mathbf{S}_{12}(\nu)\right\|_{HS}^2 + \operatorname{Tr}(\mathbf{S}_{11}(\nu))\operatorname{Tr}(\mathbf{S}_{22}(\nu))$$

The proof of Theorem 5 is based on the decomposition in Lemma 3 and is provided in supplementary information.

This estimate requires specific bias estimation techniques to develop an independence test, we will call it the *biased* estimate of the KCSD squared norm. Having the KCSD defined in an Hilbert space also enables to define similarity between two KCSD operators, so that it is possible to compare quantitatively whether different dynamical systems have similar couplings. The following theorem shows how periodograms enable to estimate the scalar product between two KCSD operators, which reflects their similarity.

**Theorem 7.** *Assume assumptions of Theorem 5 hold for two independent samples of bivariate time series* $\{(X_1(t), X_2(t))\}_{t=\ldots,-1,0,1,\ldots}$ *and* $\{(X_3(t), X_4(t))\}_{t=\ldots,-1,0,1,\ldots}$, *mapped with the same couple of reproducing kernels.*

$$\text{Then } \lim_{T \to +\infty} \mathbb{E}\langle \mathbf{P}_{12}^T(\nu), \mathbf{P}_{34}^T(\nu) \rangle_{HS} = \langle \mathbf{S}_{12}(\nu), \mathbf{S}_{34}(\nu) \rangle_{HS}, \ \nu \not\equiv 0 \pmod{1/2}$$

The proof of Theorem 7 is similar to the one of Theorem 6 provided as supplemental information.

Interestingly, this estimate of the scalar product between KCSD operators is unbiased. This comes from the assumption that the two bivariate series are independent. This provides a new opportunity to estimate the Hilbert-Schmidt norm as well, in case two independent samples of the same bivariate series are available.

**Corollary 8.** *Assume assumptions of Theorem 5 hold for the bivariate time series* $\{(X_1(t), X_2(t))\}_{t \in \mathbb{Z}}$ *and assume* $\{(\tilde{X}_1(t), \tilde{X}_2(t))\}_{t \in \mathbb{Z}}$ *an independent copy of the same time series, providing the periodogram estimates* $\mathbf{P}_{12}^T(\nu)$ *and* $\tilde{\mathbf{P}}_{12}^T(\nu)$, *respectively.*

$$\text{Then } \lim_{T \to +\infty} \mathbb{E}\langle \mathbf{P}_{12}^T(\nu), \tilde{\mathbf{P}}_{12}^T(\nu) \rangle_{HS} = \left\| \mathbf{S}_{12}(\nu) \right\|_{HS}^2, \ \nu \not\equiv 0 \pmod{1/2}$$

In many experimental settings, such as in neuroscience, it is possible to measure the same time series in several independent trials. In such a case, corollary 8 states that estimating the Hilbert-Schmidt norm of the KCSD without bias is possible using two intependent trials. We will call this estimate the *unbiased* estimate of the KCSD squared norm.

These estimate can be computed efficiently for T equispaced frequency samples using the fast Fourier transform of the centered kernel matrices of the two time series. In general, the choice of the kernel is a trade-off between the capacity to capture complex dependencies (a characteristic kernel being better in this respect), and the convergence rate of the estimate (simpler kernels related to lower order statistics usually require less samples). Related theoretical analysis can be found in [8, 12]. Unless otherwise stated, the Gaussian RBF kernel with bandwidth parameter $\sigma$, $k(x,y) = \exp(\|x-y\|^2/2\sigma^2)$, will be used as a characteristic kernel for vector spaces. Let $\mathbf{K}_{ij}$ denote the kernel matrix between the i-th and j-th time series (such that $(\mathbf{K}_{ij})_{k,l} = k(x_i(k), x_j(l))$), $\mathbf{W}$ the windowing matrix (such that $(\mathbf{W})_{k,l} = w_T(k)w_T(l)$) and $\mathbf{M}$ be the centering matrix $\mathbf{M} = \mathbf{I} - \mathbf{1}_T \mathbf{1}_T^T / T$, then we can define the windowed centered kernel matrices $\tilde{\mathbf{K}}_{ij} = (\mathbf{MK}_{ij}\mathbf{M}) \circ \mathbf{W}$. Defining the Discrete Fourier Transform matrix $\mathbf{F}$, such that $(\mathbf{F})_{k,l} = \exp(-\mathbf{i}2\pi kl/T)/\sqrt{T}$, the estimated scalar product is

$$\langle \mathbf{P}_{12}^T, \mathbf{P}_{34}^T \rangle_{\nu=(0,1,\ldots,(T-1))/T} = \|w\|^{-4} \operatorname{diag}(\mathbf{F}\tilde{\mathbf{K}}_{13}\mathbf{F}^{-1}) \circ \operatorname{diag}(\mathbf{F}^{-1}\tilde{\mathbf{K}}_{24}\mathbf{F}),$$

which can be efficiently computed using the Fast Fourier Transform ($\circ$ is the Hadamard product). The biased and unbiased squared norm estimates can be trivially retrieved from the above expression.

**Shuffling independence tests**

According to Theorem 1, pairwise independence between time series requires the cross-spectral density operator to be zero for all frequencies. We can thus test independence by testing whether the Hilbert-Schmidt norm of the operator vanishes for each frequency. We rely on Theorem 6 and Corollary 8 to compute *biased* and *unbiased* estimates of this norm. To achieve this, we generate a distribution of the Hilbert-Schmidt norm statistics under the null hypothesis by cutting the time interval in non-overlapping blocks and matching the blocks of each time series in pairs at random. Due to the central limit theorem, for a sufficiently large number of time windows, the empirical average of the statistics approaches a Gaussian distribution. We thus test whether the empirical mean differs from the one under the null distribution using a t-statistic. To prevent false positive resulting from multiple hypothesis testing, we control the Family-wise Error Rate (FWER) of the tests performed for each frequency. Following [16], we estimate a global maximum distribution on the family of t-statistics across frequencies under the null hypothesis, and use the percentile of this distribution to assess the significance of the original t-statistics.

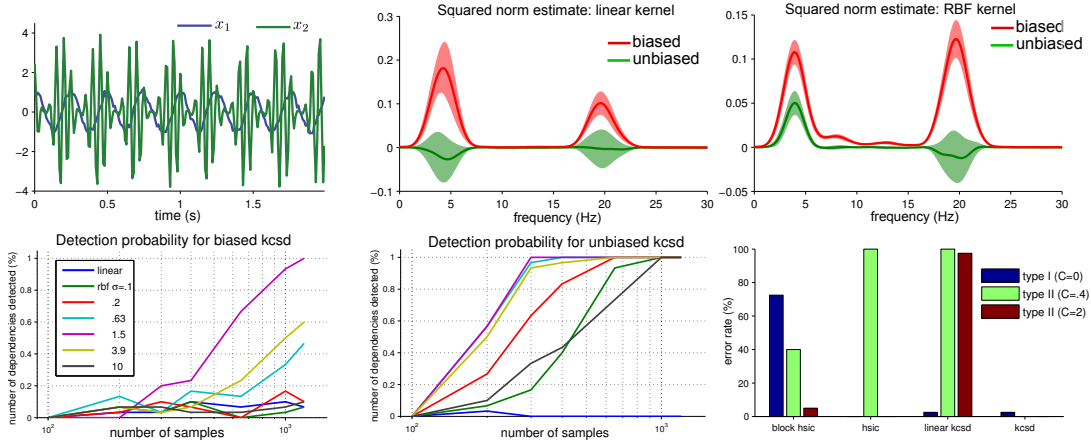

Figure 1: Results for the phase-amplitude coupling system. Top-left: example time course. Top-middle: estimate of the KCSD squared norm with a linear kernel. Top-right: estimate of the KCSD squared norm with an RBF kernel. Bottom-left: performance of the *biased kcsd* test as a function of number of samples. Bottom-middle: performance of the *unbiased kcsd* test as a function of number of samples. Bottom-right: Rate of type I and type II errors for several independence tests.

## 4 Experiments

In the following, we validate the performance of our test, called *kcsd*, on several datasets in the biased and unbiased case. There is no general time series analysis tool in the literature to compare with our approach on all these datasets. So our main source of comparison will be the HSIC test of independence (assuming data is i.i.d.). This enables us, to compare both approaches using the same kernels. For vector data, one can compare the performance of our approach with a linear dependency measure: we do this by implementing our test using a linear kernel (instead of an RBF kernel), and we call it *linear kscd*. Finally, we use the alternative approach of structured HSIC [22] by cutting the time series in time windows (using the same approach as our independence test) and considering each of them as a single multivariate sample. This will be called *block hsic*. The bandwidth of the HSIC methods is chosen proportional to the median norm of the sample points in the vector space. The p-value for all independence tests will be set to 5%.

**Phase amplitude coupling**

We first simulate a non-linear dependency between two time series by generating two oscillations at frequencies $f_1$ and $f_2$, and introducing a modulation of the amplitude of the second oscillation by the phase of the first one. This is achieved using the following discrete time equations:

$$
\begin{cases}
\varphi_1(k+1) &= \varphi_1(k) + .1\epsilon_1(k) + 2\pi f_1 T_s \\
\varphi_2(k+1) &= \varphi_2(k) + .1\epsilon_2(k) + 2\pi f_2 T_s
\end{cases}
\qquad
\begin{cases}
x_1(k) &= \cos(\varphi_1(k)) \\
x_2(k) &= (2 + C\sin\varphi_1(k))\cos(\varphi_2(k))
\end{cases}
$$

Where the $\epsilon_i$ are i.i.d normal. A simulation with $f_1 = 4Hz$ and $f_2 = 20Hz$ for a sampling frequency $1/T_s$=100Hz is plotted on Figure 1 (top-left panel). For the parameters of the periodogram, we used a window length of 50 samples (.5 s). We used a Gaussian RBF kernel to compute non-linear dependencies between the two time series after standardizing each of them (divide them by their standard deviation). The top-middle and top-right panels of Figure 1 plot the mean and standard errors of the estimate of the squared Hilbert-Schmidt norm for this system (for $C = .1$) for a linear and a Gaussian RBF kernel (with $\sigma = 1$) respectively. The bias of the first estimate appears clearly in both cases at the two power picks of the signals for the biased estimate. In the second (unbiased) estimate, the spectrum exhibits a zero mean for all but one peak (at 4Hz for the RBF kernel), which corresponds to the expected frequency of non-linear interaction between the time series. The observed negative values are also a direct consequence of the unbiased property of our estimate (Corollary 8). The influence of the bandwidth parameter of the kernel was studied in the case of weakly coupled time series ($C = .4$). The bottom left and middle panels of Figure 1 show

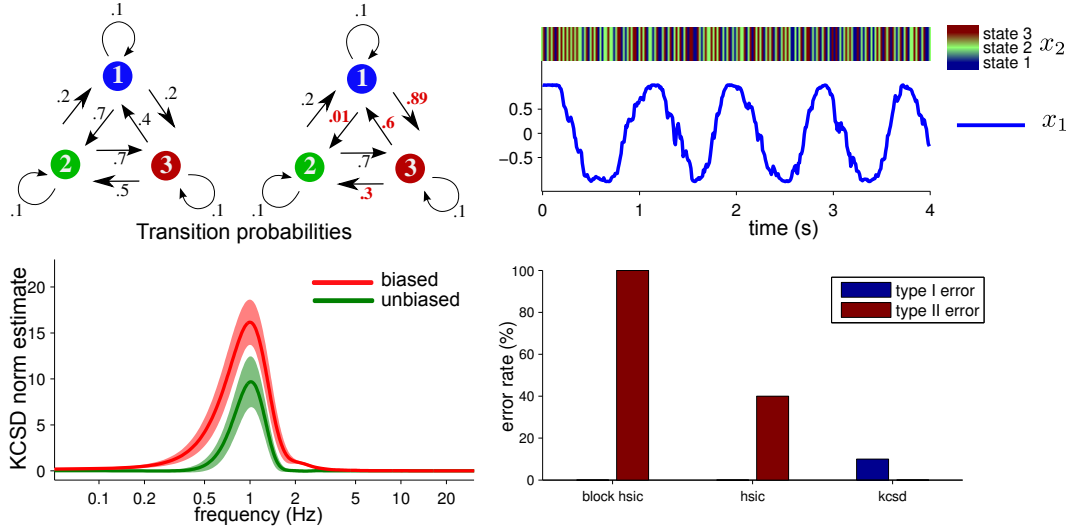

Figure 2: Markov chain dynamical system. Upper left: Markov transition probabilities, fluctuating between the values indicated in both graphs. Upper right: example of simulated time series. Bottom left: the biased and unbiased KCSD norm estimates in the frequency domain. Bottom right: type I and type II errors for *hsic* and *kcsd* tests

the influence of this parameter on the number of samples required to actually reject the null hypothesis and detect the dependency for biased and unbiased estimates respectively. It was observed that choosing an hyper-parameter close to the standard deviation of the signal (here 1.5) was an optimal strategy, and that the test relying on the unbiased estimate outperformed the biased estimate. We thus used the unbiased estimate in our subsequent analysis. The coupling parameter C was further varied to test the performance of independence tests both in case the null hypothesis of independence is true (C=0), and when it should be rejected ($C = .4$ for weak coupling, $C = 2$ for strong coupling). These two settings enable to quantify the type I and type II error of the tests, respectively. The bottom-right panel of Figure 1 reports these errors for several independence tests. Showing the superiority of our method especially for type II errors. In particular, methods based on HSIC fail to detect weak dependencies in the time series.

**Time varying Markov chain**

We now illustrate the use of our test in an hybrid setting. We generate a symbolic time series $x_2$ using the alphabet $S = [1, 2, 3]$, controlled by a scalar time series $x_1$. The coupling is achieved by modulating across time the transition probabilities of the Markov transition matrix generating the symbolic time series $x_2$ using the current value of the scalar time series $x_1$. This model is described by the following equations with $f_1 = 1Hz$.

$$\begin{cases} \varphi_1(k+1) &= \varphi_1(k) + .1\epsilon_1(k) + 2\pi f_1 T_s \\ x_1(k+1) &= \sin(\varphi_1(k+1)) \\ p(x_2(k+1) = S_i | x_2(k) = S_j) &= M_{ij} + \Delta M_{ij} x_1(k) \end{cases}$$

Since $x_1$ is bounded between -1 and 1, the Markov transition matrix fluctuates across time between two models represented Figure 2 (top-left panel). A model without these fluctuations ($\Delta M = 0$) was simulated as well to measure type I error. The time course of such an hybrid system is illustrated on the top-right panel of the same figure. In order to measure the dependency between these two time series, we use a k-spectrum kernel [14] for $x_2$ and a RBF kernel for $x_1$. For the k-spectrum kernel, we use k=2 (using k=1, i.e. counting occurrences of single symbols was less efficient) and we computed the kernel between words of 3 successive symbols of the time series. We used an RBF kernel with $\sigma = 1$, decimated the signals by a factor 2 and signals were cut in time windows of 100 samples. The biased and unbiased estimates of the KCSD norm are represented at the bottom-left of Figure 2 and show a clear peak at the modulating frequency (1Hz). The independence test results shown at the bottom-right of Figure 2 illustrate again the superiority of KCSD for type II error, whereas type I error stays in an acceptable range.

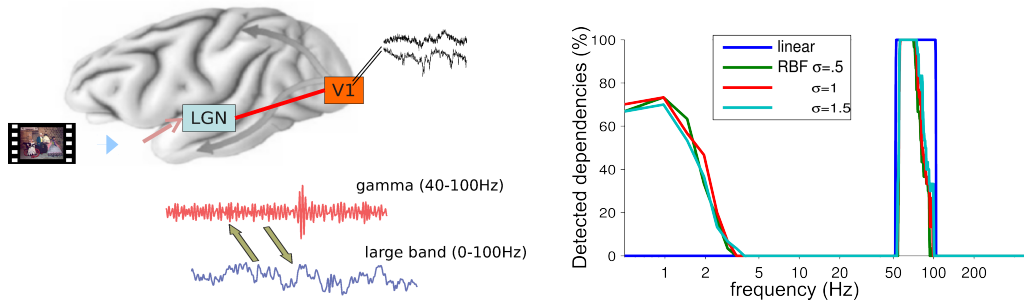

Figure 3: Left: Experimental setup of LFP recordings in anesthetized monkey during visual stimulation with a movie. Right: Proportion of detected dependencies for the *unbiased kcsd* test of interactions between Gamma band and wide band LFP for different kernels.

### Neural data: local field potentials from monkey visual cortex

We analyzed dependencies between local field potential (LFP) time series recorded in the primary visual cortex of one anesthetized monkey during visual stimulation by a commercial movie (see Figure 3 for a scheme of the experiment). LFP activity reflects the non-linear interplay between a large variety of underlying mechanisms. Here we investigate this interplay by extracting LFP activity in two frequency bands within the same electrode and quantify the non-linear interactions between them with our approach. LFPs were filtered into two frequency bands: 1/ a wide band ranging from 1 to 100Hz which contains a rich variety of rhythms and 2/ a high gamma band ranging from 60 to 100Hz which as been shown to play a role in the processing of visual information.

Both of these time series were sampled at 1000Hz. Using non-overlapping time windows of 1s points, we computed the Hilbert-Schmidt norm of the KCSD operator between gamma and large band time series originating from the same electrode. We performed statistical testing for all frequencies between 1 and 500Hz (using a Fourier transform on 2048 points). The results of the test averaged over all recording sites is plotted on Figure 3. We observe a highly reliable detection of interactions in the gamma band, using either a linear or non-linear kernel. This is due to the fact that the Gamma band LFP is a filtered version of the wide band LFP, making these signals highly correlated in the Gamma band. However, in addition to this obvious linear dependency, we observe significant interactions in the lowest frequencies (0.5-2Hz) which can not be explained by linear interaction (and is thus not detected by the linear kernel). This characteristic illustrates the non-linear interaction between the high frequency gamma rhythm and other lower frequencies of the brain electrical activity, which has been reported in other studies [21]. This also shows the interpretability of our approach as a test of non-linear dependency in the frequency domain.

## 5 Conclusion

An independence test for time series based on the concept of Kernel Cross Spectral Density estimation was introduced in this paper. It generalizes the linear approach based on the Fourier transform in several respects. First, it allows quantification of non-linear interactions for time series living in vector spaces. Moreover, it can measure dependencies between more complex objects, including sequences in an arbitrary alphabet, or graphs, as long as an appropriate positive definite kernel can be defined in the space of each time series. This paper provides asymptotic properties of the KCSD estimates, as well as an efficient approach to compute them on real data. The space of KCSD operators constitutes a very general framework to analyze dependencies in multivariate and highly structured dynamical systems. Following [13, 18], our independence test can further be combined to recent developments in kernel time series prediction techniques [20] to define general and reliable multivariate causal inference techniques.

**Acknowledgments.** MB is grateful to Dominik Janzing for fruitful discussions and advice.

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
