[Supplementary Material]

# Statistical analysis of coupled time series with Kernel Cross-Spectral Density operators.
## Supplementary information

**Michel Besserve**[*,†]       **Nikos K. Logothetis**[†]       **Bernhard Schölkopf**[*]

[*] Max Planck Institute for Intelligent Systems; [†] Max Planck Institute for Biological Cybernetics
Tübingen, Germany

## Proof of Lemma 3

Using the expression of the quadricumulant of centered variables, we show for any functions $f_1, f_3 \in \mathcal{H}_1$ and $f_2, f_4 \in \mathcal{H}_2$.

$$\left\langle f_1 \otimes f_2^*, (\mathcal{K}_{1,2,3,4})(f_3 \otimes f_4^*) \right\rangle = \mathrm{Cum}(f_1(X_1), f_2(X_2), f_3(X_3), f_4(X_4))$$
$$= \mathbb{E}\left( \left\langle X_1 \otimes X_2^*, f_1 \otimes f_2^* \right\rangle \left\langle f_3 \otimes f_4^*, X_3 \otimes X_4^* \right\rangle \right)$$
$$-\left\langle \mathbf{C}_{12}, f_1 \otimes f_2^* \right\rangle \left\langle \mathbf{C}_{34}, f_3 \otimes f_4^* \right\rangle - \left\langle \mathbf{C}_{13}, f_1 \otimes f_3^* \right\rangle \left\langle \mathbf{C}_{24}, f_2 \otimes f_4^* \right\rangle - \left\langle \mathbf{C}_{14}, f_1 \otimes f_4^* \right\rangle \left\langle \mathbf{C}_{23}, f_2 \otimes f_3^* \right\rangle$$

Then we apply this formula to the scalar product by using two orthonormal systems $\{\boldsymbol{\alpha}_i\}$ and $\{\boldsymbol{\beta}_j\}$ of $\mathcal{H}_1$ and $\mathcal{H}_2$ respectively

$$\mathbb{E}\left[ \left\langle X_1, X_3 \right\rangle \left\langle X_2, X_4 \right\rangle \right] = \mathbb{E}\left\langle X_1 \otimes X_2^*, X_3 \otimes X_4^* \right\rangle = \mathbb{E}\sum_{i,j} \left\langle X_1 \otimes X_2^*, \alpha_i \otimes \beta_j^* \right\rangle \left\langle X_3 \otimes X_4^*, \alpha_i \otimes \beta_j^* \right\rangle$$
$$= \sum_{i,j} \left\langle \alpha_i \otimes \beta_j^*, (\boldsymbol{\mathcal{K}}_{1,2,3,4})(\alpha_i \otimes \beta_j^*) \right\rangle + \sum_{i,j} \left\langle \alpha_i \otimes \beta_j^*, \boldsymbol{C}_{1,2} \right\rangle \left\langle \alpha_i \otimes \beta_j^*, \boldsymbol{C}_{3,4} \right\rangle$$
$$+ \sum_{i,j} \left\langle \alpha_i \otimes \alpha_j^*, \boldsymbol{C}_{1,3} \right\rangle \left\langle \beta_i \otimes \beta_j^*, \boldsymbol{C}_{2,4} \right\rangle + \sum_{i,j} \left\langle \alpha_i \otimes \beta_j^*, \boldsymbol{C}_{1,4} \right\rangle \left\langle \beta_i \otimes \alpha_j^*, \boldsymbol{C}_{2,3} \right\rangle$$
$$= \mathrm{Tr}\, \boldsymbol{\mathcal{K}}_{1,2,3,4} + \left\langle \boldsymbol{C}_{1,2}, \boldsymbol{C}_{3,4} \right\rangle + \mathrm{Tr}\, \boldsymbol{C}_{1,3}\, \mathrm{Tr}\, \boldsymbol{C}_{2,4} + \left\langle \boldsymbol{C}_{1,4}, \boldsymbol{C}_{3,2} \right\rangle$$

## Proof of Lemma 4

We prove this lemma for $k > 0$ (the alternative case is similar):

$$\left| \sum_t w_T(t+k)w(t) - \sum_t w_T(t)^2 \right| \leq \sum_t |w(t)| \sum_{p=0}^{k-1} |w(t+p+1) - w(t+p)|.$$

Using successively the bounded and bounded variation properties we get:

$$\left| \sum_t w_T(t+k)w(t) - \sum_t w_T(t)^2 \right| \leq (\sup |w(t)|) \sum_{p=0}^{k-1} \sum_t |w(t+p+1) - w(t+p)| \leq C|k|$$

## Riemann-Lebesgue Lemma

The following lemma is introduced to prove Theorem 6

**Lemma 10.** *[Variant of the Riemann-Lebesgue Lemma] For $\nu \neq 0 (mod\, 1)$, and $w$ Lebesgue integrable $1/T \sum_{t\in\mathbb{Z}} w(k/T) \exp(-2i\pi\nu k) \underset{T\to\infty}{\to} 0$*

*Proof.* Let us choose $\epsilon$ arbitrary small. According to the Riemann-Lebesgue Lemma, for $T > T_0$

$$\left| \int_{\mathbb{R}} w(t) \exp(-2i\pi\nu t T) dt \right| < \epsilon/2$$

Moreover, we can choose $T > T_1$ such that the piece-wise constant function $p_T(t) = w(k/T)$, $k/T \leq t < (k+1)/T$ verifies $\int |p_T(t) - w(t)| dt < \epsilon/2$ then

$$| \int (p_T(t) - w(t)) \exp(-2i\pi\nu tT) dt| < \int |p_T(t) - w(t)| < \epsilon/2$$

And thus, $| \int p_T(t) \exp(-2i\pi\nu tT) dt| < \epsilon$,

This proves that $\sum_{k=\infty}^{+\infty} w(k/T) \int_{k/T}^{(k+1)/T} \exp(-2i\pi\nu tT) dt \to 0$ Which is equivalent to

$$1/T \sum_{k=-\infty}^{+\infty} w(k/T)(\exp(-2i\pi\nu k) - \exp(-2i\pi\nu(k+1))) \to 0$$

by dividing by $(1 - \exp(-2i\pi\nu))$ we get the result

$$1/T \sum_k w(k/T) \exp(-2i\pi\nu k) \to 0$$

$\square$

**Proof of Theorem 6**

We assume that the population mean elements are used for centering (the case where empirical mean elements are used instead is asymptotically equivalent):

$$T^{-2} \mathbb{E} \Big\| \sum_{k\in\mathbb{Z}} w_T(k) \mathbf{X}_1^c(k) z^{-k} \Big\|^2 \Big\| \sum_{n\in\mathbb{Z}} w_T(n) \mathbf{X}_2^c(n) z^{-n} \Big\|^2 =$$
$$\sum_{k,n,k',n'} w_T(k) w_T(n) w_T(k') w_T(n') z^{+k-n-k'+n'} \big[ \langle \boldsymbol{C}_{12}(k-n), \boldsymbol{C}_{12}(k'-n') \rangle \dots$$
$$+ \langle \boldsymbol{C}_{12}(k-n'), \boldsymbol{C}_{12}(k'-n) \rangle + \mathrm{Tr}(\boldsymbol{C}_{11}(k-k')) \mathrm{Tr}(\boldsymbol{C}_{22}(n-n')) + \mathrm{Tr}(\boldsymbol{\mathcal{K}}(n-k, k'-k, n'-k)) \big]$$

We focus on the first term of the sum: as T grows,

$$T^{-2} \sum_{k,n,k',n'} w_T(k) w_T(n) w_T(k') w_T(n') z^{+k-n-k'+n'} \langle \boldsymbol{C}_{12}(k-n), \boldsymbol{C}_{12}(k'-n') \rangle$$

is arbitrary close to (using lemma 4 and $\sum_{k\in\mathbb{Z}} |k| \|\mathbf{C}_{12}(k)\|_{\mathrm{HS}} < +\infty$)

$$T^{-2} \sum_{k,\delta,k',\delta'} w_T(k)^2 w_T(k')^2 z^{+\delta-\delta'} \langle \boldsymbol{C}_{12}(\delta), \boldsymbol{C}_{12}(\delta') \rangle \underset{T\to+\infty}{\to} \|w\|^4 \|\boldsymbol{S}_{12}(z)\|_{HS}^2$$

For the second term of the sum: as T grows

$$T^{-2} \sum_{k,n,k',n'} w_T(k) w_T(n) w_T(k') w_T(n') z^{+k-n-k'+n'} \langle \boldsymbol{C}_{12}(k-n'), \boldsymbol{C}_{12}(k'-n)) \rangle$$

Due to the bounded variation of $w$, it is arbitrary close to

$$T^{-2} \sum_{k,k'} w_T(k)^2 w_T(k')^2 z^{+2k-2k'} \langle \sum_\delta \boldsymbol{C}_{12}(\delta) z^{-\delta}, \sum_{\delta'} \boldsymbol{C}_{12}(\delta') z^{-\delta'} \rangle \underset{T\to+\infty}{\to} 0$$

(the limit is computed using Lemma 10).

For the third term of the sum: as T grows

$$T^{-2} \sum_{k,n,k',n'} w_T(k) w_T(n) w_T(k') w_T(n') z^{+k-n-k'+n'} \mathrm{Tr}(\boldsymbol{C}_{11}(k-k')) \mathrm{Tr}(\boldsymbol{C}_{22}(n-n'))$$

is arbitrary close to (using lemma 5 and $\sum_{k=-\infty}^{+\infty} |k| \mathrm{Tr}(\mathbf{C}_{ii}(k)) < +\infty$)

$$T^{-2} \sum_{k,\delta,k',\delta'} w_T(k')^2 w_T(n')^2 z^{+\delta-\delta'} \mathrm{Tr}(\boldsymbol{C}_{11}(\delta)) \mathrm{Tr}(\boldsymbol{C}_{22}(\delta')) \underset{T\to+\infty}{\to} \|w\|^4 \mathrm{Tr}(\boldsymbol{S}_{11}(z)) \mathrm{Tr}(\boldsymbol{S}_{22}(z))$$

Finally the last term vanishes using the assumption $\sum_{k,i,j=-\infty}^{+\infty} \big| \mathrm{Tr}\, [\boldsymbol{\mathcal{K}}_{1212}(k,i,j)] \big| < +\infty$. $\square$