[Reviews · NeurIPS 2013]

Submitted by Assigned_Reviewer_2

The paper introduced a novel independency test for time series, which is based on reproducing Hilbert kernel theory (Kernel Cross Spectral Density estimation). In particular, the key object is the reproducing kernel, so the method can be applied on complex time series such as non-numerical time series. Both theoretical study and empirical results are presented.

This is certainly one of those technically dense papers. It involves several aspects of applied math in one way or another: signal processing, statistical test, RKHS, graphical models, Markov process, function operators, time series, and, of course at last, neural science. More importantly, all these techniques are involved in a non-trivial way.

The key step of using RKHS in analyzing time series is to connect the covariance matrix and reproducing kernel, and further extend to operators in functional spaces. The paper did a good job make each step work nicely together. I did not read the supplementary materials, but I think the technical content seems fine. Overall, this is a well executed paper in many aspects.

Several questions and comments:

1, the paper seems a little too crowed for the content. But given the page limit, it is a conflict. I am sure the authors will have a longer version.

2, several details:

line 91, “it’s” ? should it be “its” ?
line 95-96, “a couple of”, should it be a “coupled”?
line 290, top-left? Should it be top-right?
In Fig.1, Why the estimate of a norm can be negative? Does it come from the left hand side of Corollary 8? A bit explanation would be great.

line 418, “is in accordance with other studies”: it would be great to have relevant references here. In fact, this can be put at the very beginning of the paper as one motivation.

Summary: This paper involves many technical areas, and all these techniques are involved in a non-trivial way. Overall, this is a well executed paper in many aspects.

Submitted by Assigned_Reviewer_4

Detection of potential interdependencies between timeseries is an important problem. This paper introduces an analysis of kernel cumulant methods and via estimation of higher-order cross-spectra, the paper links to particular forms of independence testing.

There is a vast literature on higher-order methods in signal processing, from higher-order spectra to mixed norm methods of signal separation and coupling analysis. I doubt space permitted the authors to acknowledge much of this domain, but the introduction to the paper does make clear that much of the work lies in the linear coupling domain.

The derivations in the paper are sound - I was able to re-derive and follow the math. I have some minor comments and qns as below:
1) The issues surrounding *any* higher order cumulants surely don't disappear with a kernel trick : namely that very large numbers of iid samples are required for good estimation. Kurtotic cumulants in particular require very large sample sets. I can find no discussion of this.
2) Although based on generic linear models, non-Gaussian [generalized] MAR models are widely used for assessing higher-order spectra and cross-coupling, with the non-Gaussian excitation being inferred using generalizations of EM with a GMM. Links with this work?
3) The latter models link neatly with ICA style approaches, which [for certain assumed heavy-tailed density models] allow for independence to be related to negentropy and hence higher-order cumulants in the multi-var space. There are clear links with this work.
4) Fig 3. Is the blue curve under the green in the 50-100Hz region?
Summary: A fairly well-written paper, which details a more extensive theoretical treatment of kernel higher-order cross-spectra and coupling. The paper is sound in the math, and goes some way to provide an underpinning for kernel coupling approaches. The choice of real-world example does not do much to highlight the method.

Submitted by Assigned_Reviewer_5

The paper considers kernel cross-spectral density (KCSD) as a way to determine interactions among time series in a better way than methods developed under the i.i.d assumption, which is typically violated for time series data. Such a framework was originally proposed in [Besserve et al, ICASSP, 2011]. The main contribution here is to characterize cases where KCSD can be used to test independence, and to propose and study a way to estimate the properties of cross-spectral density operators from finite samples. The method is compared to the Hilbert-Schmidt Independence Criterion test under the same kernels on simulated and real neural data.

Developing independence test for time series data that are able to cope with non-linear interactions is an important area of research. The present paper provides a sound and well-motivated approach.

Many applications, however, consider a large number of time series data. Even though the proposed approach accommodate very general forms of dependencies, it is a *pair-wise* independence test, and thus suffers from the limitations inherent to pair-wise testing. It would be interesting to discuss whether the proposed method could be extended to testing simultaneously multiple time series (each being potentially multivariate).

The methodology and results depend on the choice of kernel, and various kernels might lead to contradictory conclusions. This should be discussed.

Kernel methods seem very promising candidates in capturing dependencies in time series, in the present setting and beyond. For instance, it would also be relevant to mention the recent work by Sindhwani et al, Scalable Matrix-valued Kernel Learning for High-dimensional Nonlinear Multivariate Regression and Granger Causality, UAI 2013, which uses kernel methods to capture non-linearity in causal inference, via a generalized form of sparse vector autoregression.
Summary: Developing independence tests that are tailored to time series data and can accommodate non-linearity is an important research topic. This paper proposes a sound and well-motivated approach. However, the proposed approach can only deal with pair-wise tests, while many problems involve dependence involving multiple time series, and relies on a pre-specified kernel choice.
Author Feedback

Author rebuttal: We are grateful to the reviewers for their careful reading of our manuscript and their suggestions. Responses follow.

I. Novelty and impact.
KCSD is a complex object to study and an important aspect of our contribution is to estimate its properties with good asymptotic result under mild conditions. We introduce an unbiased estimate and a statistical test using fast algorithms which are easily applicable to many datasets. Our results cannot be found elsewhere and further work can build on our mathematical treatment to assess statistical properties of kernel methods for stationary data. Most importantly, this contribution aims at bringing results from kernel methods to communities that are in important need for general time series analysis techniques with good statistical properties. Measures describing the dependency structure of the data without model assumptions, such as the linear cross-spectrum, became standard in applications such as Neurophysiology. Our approach provides a non-linear generalization of this quantity, which enables a model free statistical assessment of the dependency between time series using minimal assumptions on the system (Theorem 1, Proposition 2). We believe this measure can become a new reference in many applications related to time series. In particular, non-linear interactions are ubiquitous in brain signals and our approach provides a simple way to map these interactions in the frequency domain.

II. Links to time series models.
Reviewer 4 suggests interesting links with the use of higher order statistics in multivariate time series models and system identification techniques (in a broad sense). We included more references related to this topic by adding the following text on line 38 after “specific contexts”:
“and have been extensively used in system identification, causal inference and blind source separation (see for example [Giannakis 1989; Cardoso 1999;Hyvarinen 2009])”.
While the present paper focuses on the study of the kernel dependency measure in itself, it can be connected to time series model estimation. Indeed, most time series models rely on the assumption of i.i.d. innovations (or residuals). These assumptions are key to estimate model parameters and to validate the model. As a consequence, several methods rest on testing or maximizing independence in order to fit a model [Hyvarinen 2008;Peters]. Our independence measure, which is robust to non i.i.d. samples, can be used in similar frameworks to improve these techniques. In particular, it can be combined with recent kernel regression techniques suggested by Reviewer 5.
We added the following related sentence to line 431:
“Following [Hyvarinen 2008;Peters], our independence test can be combined to recent developments in kernel time series prediction techniques [Sindhwani 2013] to define more general and reliable multivariate causal inference techniques.”

III. Dependency between multiple time series.
We agree with Reviewer 5 that pairwise independence does not capture all the dependency structure in case more than two time series are involved. However, using the faithfulness assumption, it is possible to combine pairwise independence tests with multivariate regression techniques to fully characterize this dependency structure (see [Peters] and references therein). As mentioned in the previous paragraph, our method can thus be used to validate or fit models involving more than two time series, for example by applying it to the residuals of multivariate regressions.

IV. Choice of the kernel and connections to higher order statistics.
As mentioned by the reviewers, the choice of the kernel can affect the outcome of the analysis and can depend on the number of samples available. As mentioned in the paper, ability to detect any dependency will depend on whether the kernels are characteristic or not. However, in relation to difficulties in estimating higher order statistics, reliable estimation with a characteristic kernel might require more samples, so simpler kernels can be used first to capture the most obvious dependencies in the data. Kernel selection has been studied in a related context in [Sriperumbudur 2009;Gretton 2012].
We added this sentence on line 241:
“In general, the choice of the kernel is a trade-off between the ability to detect complex dependencies (a characteristic kernel being more sensitive), and the convergence rate of the estimate (simpler kernels related to lower order statistics usually require less samples). Related theoretical analysis can be found in [Sriperumbudur 2009;Gretton 2012].”


Detailed comments from Reviewer 2.
Regarding negative values of our estimate we added the sentence on line 296: “The observed negative values are also a direct consequence of the unbiased property of our estimate (Corollary 8).” Also, we fixed the mentioned typos (lines 91, 95 and 290). Finally, we added the reference [Whittingstal 2009] on line 418.

Reviewer 4, question 4: Yes, on Fig. 3 curves are superimposed in the gamma band.

References:
Cardoso, High-order contrasts for independent component analysis. Neu. Comput. 1999.
Giannakis et al., Identification of nonminimum phase systems using higher order statistics. IEEE TSP 1989.
Gretton et al., Optimal kernel choice for large-scale two-sample tests. NIPS 2012.
Peters et al., Causal Inference on Time Series using Structural Equation Models. arXiv.
Hyvarinen et al., Causal modeling combining instantaneous and lagged effects: an identifiable model based on non-gaussianity. ICML 2008.
Fukumizu et al., Kernel choice and classifiability for RKHS embeddings of probability distributions. NIPS 2009.
Whittingstall et al., Frequency-band coupling in surface EEG reflects spiking activity in monkey visual cortex. Neuron 2009.